# Using natural language processing to extract structured epilepsy data from unstructured clinic letters: development and validation of the ExECT (extraction of epilepsy clinical text) system

Beata Fonferko-Shadrach,[1] Arron S Lacey,[1,2] Angus Roberts,[3] Ashley Akbari,[2] Simon Thompson,[2] David V Ford,[2] Ronan A Lyons,[2] Mark I Rees,[1,4] William Owen Pickrell[1]

BF-S and ASL contributed equally.

For numbered affiliations see end of article.

**Correspondence to**
Dr William Owen Pickrell;
w.o.pickrell@swansea.ac.uk

## ABSTRACT

**Objective** Routinely collected healthcare data are a powerful research resource but often lack detailed disease-specific information that is collected in clinical free text, for example, clinic letters. We aim to use natural language processing techniques to extract detailed clinical information from epilepsy clinic letters to enrich routinely collected data.

**Design** We used the general architecture for text engineering (GATE) framework to build an information extraction system, ExECT (extraction of epilepsy clinical text), combining rule-based and statistical techniques. We extracted nine categories of epilepsy information in addition to clinic date and date of birth across 200 clinic letters. We compared the results of our algorithm with a manual review of the letters by an epilepsy clinician.

**Setting** De-identified and pseudonymised epilepsy clinic letters from a Health Board serving half a million residents in Wales, UK.

**Results** We identified 1925 items of information with overall precision, recall and F1 score of 91.4%, 81.4% and 86.1%, respectively. Precision and recall for epilepsy-specific categories were: epilepsy diagnosis (88.1%, 89.0%), epilepsy type (89.8%, 79.8%), focal seizures (96.2%, 69.7%), generalised seizures (88.8%, 52.3%), seizure frequency (86.3%–53.6%), medication (96.1%, 94.0%), CT (55.6%, 58.8%), MRI (82.4%, 68.8%) and electroencephalogram (81.5%, 75.3%).

**Conclusions** We have built an automated clinical text extraction system that can accurately extract epilepsy information from free text in clinic letters. This can enhance routinely collected data for research in the UK. The information extracted with ExECT such as epilepsy type, seizure frequency and neurological investigations are often missing from routinely collected data. We propose that our algorithm can bridge this data gap enabling further epilepsy research opportunities. While many of the rules in our pipeline were tailored to extract epilepsy specific information, our methods can be applied to other diseases and also can be used in clinical practice to record patient information in a structured manner.

## Strengths and limitations of this study

► This study presents a novel method to automatically extract detailed, structured epilepsy information from unstructured clinic letters.
► The method is based on open-source natural language processing technology.
► The performance was validated using 200 previously unseen epilepsy and general neurology clinic letters.
► The generalisability of the algorithm to population-level data and other diseases is limited at present but is possible with further work.

## INTRODUCTION

Epilepsy is a common neurological disease with significant co-morbidity. Although advances have been made in understanding the aetiology, treatment and co-morbidity of epilepsy, significant uncertainties still exist. Research using routinely collected data offers an opportunity to explore these uncertainties. Recent studies have shown, for example, an increased onset of psychiatric disorders and suicide before and after epilepsy diagnosis,[1] no association between anti-epileptic drug use during pregnancy and stillbirth,[2] and an increased risk of premature mortality in people with epilepsy.[3]

Epilepsy research using routinely collected data currently tends to use sources such as primary care health records or hospital discharge summaries. The main disadvantage of these sources is that they do not contain detailed epilepsy information, for example, epilepsy subtype/syndrome, epilepsy cause, seizure type or investigation results. This limits the quality and type of epilepsy research questions that can be answered successfully.

Almost all patient encounters with hospital specialists in the UK are documented in clinic letters to primary care doctors, other healthcare professionals and patients. Clinic letters have been written electronically for decades and offer a wealth of disease-specific information to enhance routinely collected data for research. Although detailed disease (epilepsy) information is found in clinic letters, they are usually written in an unstructured or semi-structured format, making it difficult to automatically extract useful information.

Natural language processing (NLP) technology can be used to analyse human language and offers a potential solution for automated information extraction from unstructured letters.[4] NLP is increasingly being used for healthcare information extraction applications; for example, to extract symptoms of severe mental illness and adverse drug events from psychiatric health records,[5 6] to identify patients with non-epileptic seizures[7] and for the early identification of patients with multiple sclerosis.[8]

In this project, our objective was to develop and validate an NLP application to extract detailed epilepsy information from unstructured clinic letters, with the primary aim of using this information to enhance epilepsy research using routinely collected data.

## MATERIALS AND METHODS
### Study population
We used manually de-identified and pseudonymised hospital clinic letters to build and test the algorithm. The letters were provided by the paediatric and neurology departments of a local general hospital. Members of the clinical team manually changed patient details, clinician details as well as the names occurring within the text before the letters were available to researchers. We used 40 letters for training purposes to build rule sets, and a validation set of 200 letters to test the accuracy of the algorithm. The training set was randomly selected and included 24 adult (16 epilepsy, 8 general neurology) and 16 paediatric neurology letters. The validation set contained letters from various outpatient clinics (145

adult epilepsy, 37 paediatric epilepsy and 18 general neurology) from new patient and follow-up appointments, written by eight different clinicians.

### Algorithm construction
We used the general architecture for text engineering (GATE) framework with its biomedical named entity linking pipeline (Bio-YODIE) (figure 1).[9] We created an automated clinical text extraction system for epilepsy, ExECT (extraction of epilepsy clinical text), which used Bio-YODIE and our own customisations to map clinical terms to Unified Medical Language System (UMLS) concepts.[10] The UMLS is a set of files and software, developed by the US National Library of Medicine, which combines information from over 200 health vocabularies with over 3.6 million concepts and 13.9 million unique concept names.[11] UMLS uses concept unique identifiers (CUIs) to identify senses (or concepts) associated with words and terms.[12] Bio-YODIE applies several strategies to assign the correct UMLS sense to terms in the text, and, where necessary, disambiguates against several possible meanings for the same term. These strategies include term frequency, patterns of co-occurrence with other terms and measures of context similarity.

We supplemented Bio-YODIE's UMLS lookups with a set of custom gazetteers (native dictionaries used within GATE). We used some custom gazetteers to embed context into extracted UMLS concepts, that is, phrases to determine present, past or future tense, or terms to describe levels of certainty expressed in clinical opinion. For example, in the phrase '… could be consistent with simple partial seizures', *simple partial seizure*s are annotated with a certainty level indicated by the word 'could'. We used five levels of certainty ranging from 1 (definitely not) to 5 (definitely) (see table 1 for more details). Variables with certainty levels 4 or 5 were considered to be positive findings and those with levels 1–3 to be negative findings. We used other gazetteers for specific vocabulary or colloquial terminology used by patients and clinicians when describing symptoms. Some were necessary to deal with the rigidity of the UMLS terminology, especially in

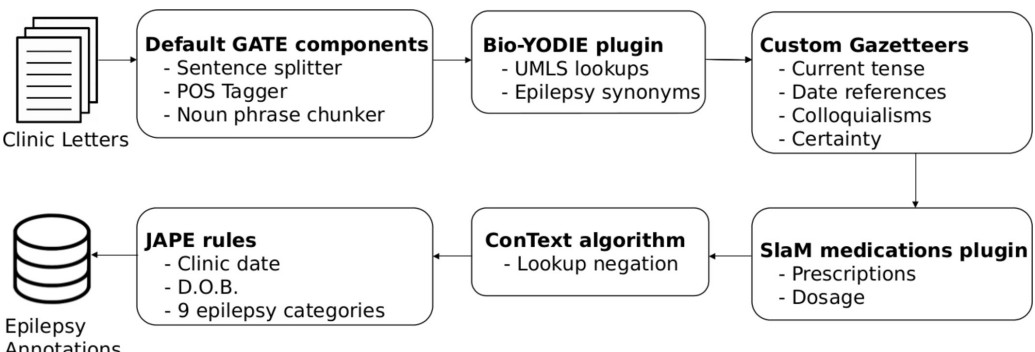

**Figure 1** ExECT pipeline to extract clinic date, date of birth and nine categories of epilepsy information from clinic letters. We used GATE architecture, modified versions of the Bio-YODIE, SLaM and ConText plugins with custom dictionaries and JAPE (Java Annotation Patterns Engine) rules. Bio-YODIE, biomedical named entity linking pipeline; POS Tagger, Part-Of-Speech Tagger; D.O.B, date of birth; ExECT, extraction of epilepsy clinical text; GATE, general architecture for text engineering; SLAM, South London and Maudsley.

**Table 1** Details on the categories of extracted information and criteria for manual review which were used as algorithm development guidelines

| Category | Details |
|---|---|
| Clinic date | The date the patient visited the clinic. |
| Date of birth | The patient's date of birth. |
| Epilepsy diagnosis | Items of information which confirmed a diagnosis of epilepsy, for example, 'this lady has a diagnosis of focal epilepsy' or '… has recurrent unprovoked generalised tonic-clonic seizures'. We specified that the epilepsy diagnosis must be attributable to the patient (eg, not a family member); and did not include items of information that described epilepsy clinic attendance, or a discussion about epilepsy in general, as confirmation of an epilepsy diagnosis. Only epilepsy diagnosis with a certainty level 4 or 5 was considered to be a true positive. |
| Epilepsy type | Whether the patient had focal or generalised epilepsy or an epilepsy syndrome where epilepsy type could be inferred. For example, *generalised epilepsy* if the letter confirmed juvenile myoclonic epilepsy. We based this information on the UMLS CUI extracted with the epilepsy diagnosis information. We only used explicit mentions of epilepsy types or syndromes within the clinic letters, and did not use other information, such as seizure type or investigation results, to infer the epilepsy type. Only an epilepsy type with a certainty level 4 or 5 was considered to be a true positive. |
| Seizure type | Specific seizure types, for example, 'focal motor seizures' or 'absence seizures'. We categorised the seizure type into focal seizures or generalised seizures at the validation stage. Only a seizure type with a certainty level 4 or 5 was considered to be a true positive. |
| Seizure frequency | The number of seizures in a specific time period, for example, 'two seizures per day', 'seven seizures in a year' or 'seizure-free since last seen in clinic.' |
| Medication | An identifiable drug name with a quantity and frequency, for example, 'Lamotrigine 250 mg bd'. |
| Investigation | The type of investigation and classification of results (normal or abnormal). We used UMLS CUI codes to assign a normal/abnormal value to investigation results, using the simplified abnormal outcomes gazetteers. We categorised the investigation results into CT, MRI and EEG results at the validation stage. |
| Levels of certainty | Not a category in itself, but an annotation qualifier addressing the uncertainty of diagnosis expressed in clinic letters. We defined five levels of certainty: (1) (no diagnosis), for example, 'epilepsy has been ruled out'; (2) (unlikely diagnosis), for example, 'I doubt that these episodes are epileptic in nature'; (3) (uncertain diagnosis), for example, 'it is possible that these are focal motor seizures'; (4) (likely diagnosis), for example, 'the impression is that this is JME' and (5) (definite diagnosis), for example, 'this patient is having complex partial seizures'. We applied these certainty levels to epilepsy diagnosis, epilepsy type and seizure type. |

relation to investigation findings, such as electroencephalogram (EEG) and MRI results. For example, the EEG abnormality indicated in the phrase: 'EEG with spike and wave activity' would not be matched with a UMLS lookup but our EEG results gazetteer would annotate 'spike and wave' as an abnormal EEG result, assigning it an UMLS CUI for an abnormal EEG outcome.

We used and customised the South London and Maudsley (SLaM) GATE application to extract prescription information.[13] We deployed the ConText algorithm to detect negation of extracted terms, that is, 'this person does *not* have epilepsy' and to identify normal test results, such as in 'The EEG was not abnormal'.[14] Finally, we used the JAPE scripting language to define rules based on varying combinations of UMLS and custom lookups to extract eight broad categories of information (see table 1 for more information). In total, we created 46 separate gazetteers and over 89 JAPE rule files in order to annotate the variables of interest, establish context and to remove certain annotations from the output.

An illustration of the ExECT GATE pipeline used to extract all items of interest is shown in figure 1. The source code is available at https://github.com/arronlacey/ExECT. ExECT was built using GATE V.8.4.1.

### Measuring performance

We ran ExECT on a validation set of 200 previously unseen, de-identified, clinic letters. We compared the items of information extracted by ExECT with those extracted by manual review. The review was performed by an epilepsy clinician (WOP) who was blinded to the algorithm results until the review was complete. We used pre-defined criteria for the manual review of information items (see table 1). The core research team (BF-S, ASL and WOP) reviewed every disagreement between the manual review and ExECT, and a consensus was obtained from the group on the correct annotation based on our pre-defined guidelines (see table 1). We measured performance on both a *per item* and a *per letter* basis.

The *per item* test compared every mention of an information item in each letter. Frequently, there were several items in a particular category. For example, a letter could contain the following phrases, all of which confirm an epilepsy diagnosis: 'diagnosis: temporal lobe epilepsy'

'… frequent complex partial seizures consistent with a diagnosis of temporal lobe epilepsy…' and 'Given that X has temporal lobe epilepsy, the best treatment is…'

The main purpose of ExECT is to enrich routinely collected data sets with epilepsy information and for this purpose a *per letter* score is potentially a more useful measure of its performance. For example, a letter may confirm temporal lobe epilepsy three times but only one mention of temporal lobe epilepsy is required to correctly classify that person's epilepsy. In this context, extracting only one mention of temporal lobe epilepsy is as useful as extracting all three. In the *per letter* test, we, therefore, aggregated multiple mentions within a category in each letter to a binary decision based on ExECT's ability to extract at least one true positive mention. In the above example, if ExECT had only correctly identified one of the three mentions of temporal lobe epilepsy, we would have scored it as having a recall of 100% on a *per letter* basis but only 33% (1/3) on a *per item* basis. We used a similar approach with seizure frequency, clinic date and date of birth; multiple mentions were counted in a *per item*, but only one true mention (in the absence of contradictory information) was considered to give a true positive result in the *per letter* method. For the medication annotation, in the *per letter* approach, only a full list of the drugs prescribed with the respective doses was considered to be a positive outcome.

### Patient and public involvement

This research was carried out without specific patient or public involvement in the design or interpretation of results. Patients and members of the public did not contribute to the writing or editing of this manuscript.

### Analysis and statistical tests

We used *precision, recall* and *F1 score* to measure the accuracy of ExECT. Precision is defined as the proportion of the instances extracted by the algorithm which are true, recall is the proportion of true instances extracted by the algorithm and F1 score is the unweighted harmonic mean of precision and recall: $(2 \times \text{precision} \times \text{recall})/(\text{precision} + \text{recall})$.

### RESULTS

We identified 1925 items in 11 categories across 200 letters. See table 2 for a summary of the performance of ExECT in identifying these items of information and table 3 for the evaluation of the results.

### DISCUSSION

We developed an application capable of extracting a range of detailed epilepsy information from unstructured epilepsy and general neurology clinic letters, in order to enrich routinely collected data for research. ExECT reliably extracted epilepsy information from 200 clinic letters, written by different clinicians, with an overall precision, recall and F1 score of 91%, 81% and 86%, respectively, on a *per item* basis. ExECT performed best in extracting clinic date and date of birth (F1 scores of 98% and 99%) given that these fields consist of fixed format dates which are easier to extract. In terms of epilepsy-specific information, ExECT performed best for medication (F1=95%), epilepsy diagnosis (89%), epilepsy type (85%) and focal seizure types (81%). These items are frequently mentioned and presented in a relatively standard format,

**Table 2** Information extracted from 200 epilepsy clinic letters: number of items and number of letters with items extracted.

**Information extracted from 200 epilepsy clinic letters**

| Variable | All items extracted | | | | | Number of letters with items extracted | | | | |
|---|---|---|---|---|---|---|---|---|---|---|
| | Clinician | Algorithm | TP | FP | FN | Clinician | Algorithm | TP | FP | FN |
| Clinic date | 191 | 188 | 186 | 2 | 5 | 186 | 181 | 181 | 0 | 5 |
| Date of birth | 201 | 197 | 197 | 0 | 4 | 199 | 195 | 195 | 0 | 4 |
| Epilepsy diagnosis | 383 | 387 | 341 | 46 | 42 | 150 | 152 | 143 | 9 | 7 |
| Epilepsy type | 89 | 79 | 71 | 8 | 18 | 70 | 67 | 61 | 6 | 9 |
| Focal seizures | 145 | 105 | 101 | 4 | 44 | 71 | 61 | 59 | 2 | 12 |
| Generalised seizures | 151 | 89 | 79 | 10 | 72 | 76 | 58 | 52 | 6 | 24 |
| Seizure frequency | 153 | 95 | 82 | 13 | 71 | 119 | 77 | 71 | 6 | 48 |
| Medication | 316 | 309 | 297 | 12 | 19 | 157 | 145 | 143 | 2 | 14 |
| CT scan | 17 | 18 | 10 | 8 | 7 | 16 | 13 | 10 | 3 | 6 |
| MRI scan | 109 | 91 | 75 | 16 | 34 | 66 | 60 | 52 | 8 | 14 |
| EEG | 170 | 157 | 128 | 29 | 42 | 79 | 82 | 71 | 11 | 8 |
| All | 1925 | 1715 | 1567 | 148 | 358 | 1189 | 1091 | 1038 | 53 | 151 |

EEG, electroencephalogram; ExECT, extraction of epilepsy clinical text; FN, false negative (an item annotated by the clinician but not annotated by ExECT); FP, false positive (an item annotated by ExECT but not annotated by the clinician); TP, true positive (an item annotated by the clinician and ExECT).

**Table 3** The *per item* and *per letter* accuracy of ExECT when extracting epilepsy information from a validation set of 200 clinic letters, where an information item is defined as a single item in any category identified by the human annotator (see the Materials and methods section for more details)

| Variable | 200 letters—results *per item* | | | | 200 letters—results *per letter* | | | |
| --- | --- | --- | --- | --- | --- | --- | --- | --- |
| | Number of information items identified by clinician | Precision % | Recall % | F1 score % | Number of letters containing information items identified by clinician | Precision % | Recall % | F1 score % |
| Clinic date | 191 | 98.9 | 97.4 | 98.2 | 186 | 100.0 | 97.3 | 98.6 |
| Date of birth | 201 | 100.0 | 98.0 | 99.0 | 199 | 100.0 | 98.0 | 99.0 |
| Epilepsy diagnosis | 383 | 88.1 | 89.0 | 88.6 | 150 | 94.1 | 95.3 | 94.7 |
| Epilepsy type | 89 | 89.9 | 79.8 | 84.5 | 70 | 91.0 | 87.1 | 89.1 |
| Focal seizures | 145 | 96.2 | 69.7 | 80.8 | 71 | 96.7 | 83.1 | 89.4 |
| Generalised seizures | 151 | 88.8 | 52.3 | 65.8 | 76 | 89.7 | 68.4 | 77.6 |
| Seizure frequency | 153 | 86.3 | 53.6 | 66.1 | 119 | 92.2 | 59.7 | 72.4 |
| Medication | 316 | 96.1 | 94.0 | 95.0 | 157 | 98.6 | 91.1 | 94.7 |
| CT scan | 17 | 55.6 | 58.8 | 57.1 | 16 | 76.9 | 62.5 | 69.0 |
| MRI scan | 109 | 82.4 | 68.8 | 75.0 | 66 | 86.7 | 78.8 | 82.5 |
| EEG | 170 | 81.5 | 75.3 | 78.3 | 79 | 86.6 | 89.9 | 88.2 |
| All | 1925 | 91.4 | 81.4 | 86.1 | 1189 | 95.1 | 87.3 | 91.1 |

EEG, electroencephalogram; ExECT, extraction of epilepsy clinical text.

for example, medication is usually stated as *drug name-strength-unit-frequency*, and diagnosis appears at the top of letters in structured lists or in text with clear references to the patient.

ExECT was less accurate in identifying CT (F1=57%), MRI (75%) and EEG results (78%), seizure frequency (66%) and generalised seizure terms (66%). These items occasionally did not map completely to UMLS terms and had a more varied format in the clinic letters. For example, UMLS contains terms such as 'EEG with irregular generalised spike and wave complexes'; however, there were often a variety of words between EEG and the associated result, for example, 'EEG was found to show generalised spike and wave complexes'. Consequently, we created custom gazetteers that map to specific terms such as 'spike and wave' or 'EEG' and wrote JAPE rules to associate these terms with the EEG term to improve the performance of our algorithm. While this approach allows for variations seen in our training set, previously unseen variations in the validation set could not be captured. Similarly, the reporting of seizure frequency is highly varied, for example, 'she had five seizures since March last year' or 'one or two focal seizures every evening'.

We achieved higher scores for precision, recall and F1 score (95%, 87% and 91%, respectively) on a *per letter* basis. A lower recall rate for medication on a *per letter* basis was due to the scoring method, where only a complete list of all medications was considered a true positive result. For example, if one medication was missing out of a list of four, this would be a negative result on a *per letter* basis. This lead to an increase in the false negative scores as compared with a *per item* approach. We propose that a *per letter* measure for categories containing multiple mentions, such as confirmation of epilepsy, provides a practical way to summarise information from clinic letters. Additionally, a *per person* measure (results summarised over several letters) could be used to determine epilepsy status as there will normally be several letters per person over a period of time.

### Strengths

We used a gold standard data set of de-identified clinic letters to build and test ExECT, from which we accurately extracted novel epilepsy information for research. We can now iteratively develop ExECT over larger sets of clinic letters and use it to extract detailed epilepsy information for research on a population-level basis. We can also develop our algorithm for other diseases and potential clinical applications, for example, efficiently extracting relevant clinical information from historical letters to aid clinicians. Our system uses UMLS terminologies including the ability to map findings to CUI codes. This can be powerful in curating structured data sets that can be easily linked with high agreement to other coding systems, for example, SNOMED-CT.[15]

We used the open source GATE framework to develop our algorithm which provides useful built-in applications and user-developed plugins for NLP such as Bio-YODIE and the SLaM medication application. This undoubtedly made the process easier and quicker than other potential

methods and provided a useful model for future similar information extraction applications.

## Weaknesses

We used a relatively small number of letters sourced from one health board. Abertawe Bro Morganwg University Health Board is responsible for planning and providing healthcare services to approximately half a million people in southwest Wales. This limited the number of writing styles and letter structures available to validate our algorithm, given that manually de-identifying letters was resource intensive. The generalisability of our algorithm may, therefore, be limited. However, we have made efforts to extract information from the main body of text within clinic letters rather than relying on the letter structure alone.

It is difficult to account for the variability of the language used to express patient information in clinic letters. Some items of information such as seizure frequency and investigations require many complex rules where patterns are hard to predict. Further work could be focused on employing machine learning methods to compliment a rule-based approach; however, this would require a significant amount of time to annotate the large amount of documents required for such a task. All disagreements between ExECT and manual annotation were reviewed by the research team as a whole but we only used one clinician to review the letters, which might have added bias to how the validation set was annotated.

## Comparison with other studies

NLP is being increasingly used for clinical information extraction purposes.[4] The i2b2 project used Apache clinical Text Analysis and Knowledge Extraction System and Health Information Text Extraction to extract the following phenotypes with positive predictive value (precision) and sensitivity (recall): Crohn's disease (98%, 64%), ulcerative colitis (97%, 68%), multiple sclerosis (MS) (94%, 68%) and rheumatoid arthritis (89%, 56%).[16] A recent study on patients with MS, identified from electronic healthcare records, used NLP techniques to extract MS-specific attributes with high positive predictive value and sensitivity, namely, Expanded Disability Status Scale (97%, 89%), Timed 25 Foot Walk (93%, 87%), MS subtype (92%, 74%) and age of onset (77%, 64%).[17] A study used clinic letters (available at www.mtsamples.com) to determine whether sentences containing disease and procedure information were attributable to a family member using the BioMedICUS NLP system. This achieved an overall precision, recall and F1 score of 91%, 94% and 92%, respectively.[18]

To our knowledge, there are only a few published studies on clinical epilepsy information extraction systems. Cui *et al* developed the rule-based epilepsy data extraction and annotation (EpiDEA) system, which extracts epilepsy information from epilepsy monitoring unit discharge summaries. EpiDEA achieved an overall precision, recall and F1 score of 94%, 84% and 89%, respectively, when extracting EEG pattern, past medications and current medication from 104 discharge summaries from Cleveland, Ohio, USA.[19] Cui *et al* also developed the rule-based phenotype extraction in epilepsy (PEEP) pipeline.[20] PEEP extracted the epileptogenic zone, seizure semiology, lateralising sign, interictal and ictal EEG pattern with an overall precision, recall and F1 score of 93%, 93% and 92%, respectively, in a validation set of 262 epilepsy monitoring unit discharge summaries from Cleveland, Ohio, USA. Sullivan *et al* used a machine-based learning NLP pipeline to identify a rare epilepsy syndrome from discharge summaries and EEG reports in Phoenix, Arizona, USA and achieved a precision, recall and F1 score of 77%, 67% and 71%, respectively.[21] The majority of these studies used discharge letters that are generally more structured than the clinic letters used in our study, which presents a greater challenge for NLP application.

## CONCLUSION

Using the GATE framework and the existing applications, we have developed an automated clinical-text extraction system, ExECT, which can accurately extract epilepsy information from free text in clinic letters. This can enhance routinely collected data for epilepsy research in the UK. The types of information extracted using our algorithm such as epilepsy type, seizure frequency and neurological investigation results are often missing from routinely collected data. We propose that our algorithm can be used to fill this data gap, enabling further epilepsy research opportunities. While many of the rules in our pipeline were tailored to extracting epilepsy specific information, the methods employed could be generalised to other disease areas and used in clinical practice to record patient information in a structured manner.

## Future work

We are developing ExECT to extract other epilepsy variables including age of seizure onset and co-morbidities. In addition, we aim to deploy ExECT in hospital information systems to enhance the availability of structured clinical data for clinicians.

**Author affiliations**
[1]Neurology and Molecular Neuroscience Group, Institute of Life Science, Swansea University Medical School, Swansea University, Swansea, UK
[2]Health Data Research UK, Data Science Building, Swansea University Medical School, Swansea University, Swansea, UK
[3]Institute of Psychiatry, Psychology and Neuroscience, King's College London, London, UK
[4]Faculty of Medicine and Health, University of Sydney, Sydney, Australia

**Contributors** WOP, ASL and BF-S were responsible for study design. ASL and BF-S developed the platform with assistance from AR and WOP. WOP, ASL and BF-S validated the algorithm and drafted the initial manuscript. AA, ST, DVF, RAL and MIR provided senior support and supervision, secured the funding and research infrastructure for the project. All authors reviewed and critically revised the manuscript.

**Funding** We acknowledge the support from the Farr Institute @ CIPHER. The Farr Institute @ CIPHER is supported by a 10-funder consortium: Arthritis Research UK, the British Heart Foundation, Cancer Research UK, The Economic and Social Research Council, The Engineering and Physical Sciences Research Council, The Medical Research Council, The National Institute of Health Research, The Health

and Care Research Wales (Welsh Assembly Government), The Chief Scientist Office (Scottish Government Health Directorates) and The Wellcome Trust (MRC Grant No: MR/K006525/1). We also acknowledge the support from the Brain Repair and Intracranial Neurotherapeutics (BRAIN) Unit which is funded by the Health and Care Research Wales. The work has also been supported by the Academy of Medical Sciences (Starter grant for medical sciences, WOP), The Wellcome Trust, The Medical Research Council, British Heart Foundation, Arthritis Research UK, and the Royal College of Physicians and Diabetes UK (SGL016\1069). AR was supported by the National Institute for Health Research (NIHR). This paper represents independent research partly funded by the NIHR Biomedical Research Centre at South London, the Maudsley NHS Foundation Trust and King's College London. The views expressed are those of the author(s) and not necessarily those of the NHS, the NIHR or the Department of Health and Social Care.

**Competing interests** WOP has receivedclinical research fellow salary support in the form of an unrestricted grantfrom UCB Pharma and has undertaken work commissioned by the biopharmaceuticalcompany. We confirm that we have read the Journal's position on issues involvedin ethical publication and affirm that this report is consistent with thoseguidelines .

**Patient consent for publication** Not required.

**Ethics approval** This research was conducted with anonymised and de-identified routinely collected clinic letters and therefore specific ethical approval was not required.

**Provenance and peer review** Not commissioned; externally peer reviewed.

**Data sharing statement** We do not have a data sharing agreement for the data, however, we are exploring ways of obtaining patient consent and endeavour to produce a minimum dataset for cross-platform testing.

**Author note** MIR and WOP were equal senior authors

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
