## [Reviewer comments · BMJ Open]

This paper was submitted to a another journal from BMJ but declined for publication following peer review. The authors addressed the reviewers' comments and submitted the revised paper to BMJ Open. The paper was subsequently accepted for publication at BMJ Open.

(This paper received three reviews from its previous journal but only two reviewers agreed to published their review.)

ARTICLE DETAILS

TITLE (PROVISIONAL)	Using natural language processing to extract structured epilepsy data from unstructured clinic letters: development and validation of the ExECT (Extraction of Epilepsy Clinical Text) system
AUTHORS	Fonferko-Shadrach, Beata; Lacey, Arron; Roberts, Angus; Akbari, Ashley; Thompson, Simon; Ford, David; Lyons, Ronan; Rees, Mark; Pickrell, W. Owen

VERSION 1 – REVIEW

REVIEWER	Saeed Hassanpour Dartmouth College, USA
REVIEW RETURNED	19-May-2018

GENERAL COMMENTS	The authors present an NLP method for extracting epilepsy data from epilepsy and neurology clinic letters. The submission needs very minor revision before it would be considered for acceptance. The authors stated they used the Context algorithm to detect negation and gave an example of a negative diagnosis, but there was no further discussion of how this played a role if any in the results. In Table 1, the authors define 5 levels of certainty for "Epilepsy diagnosis", but again it is unclear how this factors into the results. In the results, this is apparently analogous to what is presented as "Epilepsy confirmed". There needs to be more clarity across the submission on how the above items interrelate. A very brief explanation of what was done to deidentify the letters (what information was removed/modified manually) might be helpful to some readers. Table 2 is vague on the meaning of "Number of information items" and "Number of letters containing information items". Based on the caption, I am assuming these are the numbers labeled by the algorithm, but this should be made more clear. I'd suggest also including the numbers identified by the clinician in this table or a separate table that perhaps also includes numbers for TP, FP, FN.
---

REVIEWER	German Rigau Univ Basque Country UPV EHU
REVIEW RETURNED	24-Aug-2018

GENERAL COMMENTS	This paper describes a knowledge-based Information Extraction system for detecting epilepsy information in anonymised clinical notes. The system uses the open source GATE framework to build an application to extract epilepsy information from clinic letters. It makes use of the Bio-YODIE GATE plugin as well as the South London and Maudsley medication application. 40 documents have been used for development and 200 for test. The source code is available. Although interesting, the paper does not provide information on the time and effort invested in the development of the adapted system. How was the procedure for developing the final system? Which results obtain your final system on the development documents? How much code, rules, etc. have been necessary to obtain the final results? Did you perform any validation on the annotated data? Did you develop annotation guidelines for the task? Which is the Inter-Tagger Agreement between expert annotators? Is it far from the system performance? For comparing the new system to previous ones, why not applying your system to discharge letters instead of clinic letters? Moreover, no error analysis is included. Where the system is still failing? Why? The same problems in both the development and test documents? Is your system not producing false positives? No direct comparison on the same data is reported since it seems that there is no public available epilepsy documents. Is it not possible to release the annotated documents? How can we establish a fair comparison between existing systems? There are no other available systems for processing your documents? EpiDEA, PEEP, etc. are not available? What do you plan as a future work? What and how do you plan to do next? Minor comments: There are two possible papers referring to Cui et al. in section "Comparison to other studies" which refers to papers [20] and [21].
---

VERSION 1 – AUTHOR RESPONSE

We would like to thank the reviewers for their comments and for pointing out issues where we had not provided enough detail or were not sufficiently clear. We have addressed each of the comments below and hope that this is sufficient to address their concerns. We have revised the manuscript highlighting the changes to the original text in red, and we trust that we now present a much improved paper.

Reviewer 1

The authors stated they used the Context algorithm to detect negation and gave an example of a negative diagnosis, but there was no further discussion of how this played a role if any in the results.

We were only interested in a confirmed diagnosis of epilepsy such as “this person has epilepsy” or “this person has focal seizures” in this information extraction application. Refuted diagnoses, as in “this person does not have epilepsy” were not extracted as a separate category but were treated as the true negatives for the “epilepsy diagnosis” variable (and not included in the calculation of precision or recall). We have expanded the methods section to clarify this (4th paragraph).

In Table 1, the authors define 5 levels of certainty for "Epilepsy diagnosis", but again it is unclear how this factors into the results.

Levels of certainty were assigned to all diagnostic statements, based on phrases expressing clinical opinion. Our analysis was based on annotations with the certainty level 4 and above i.e. only "Epilepsy diagnosis" with a certainty level 4 or 5 was determined as a true positive result. Please see the methods section 3rd paragraph.

In the results, this is apparently analogous to what is presented as "Epilepsy confirmed". There needs to be more clarity across the submission on how the above items interrelate.

We apologise for the ambiguity here. Epilepsy diagnosis is epilepsy confirmed, and these are diagnoses with a certainty level of 4 and above – we have changed all statements in the paper to be consistent with "epilepsy diagnosis".

A very brief explanation of what was done to deidentify the letters (what information was removed/modified manually) might be helpful to some readers.

We have added some further information on the de-identification process in the methods section (1st paragraph).

Table 2 is vague on the meaning of "Number of information items" and "Number of letters containing information items". Based on the caption, I am assuming these are the numbers labelled by the algorithm, but this should be made clearer.

We have changed and highlighted the headings in Table 2 (Table 3 in the revised manuscript) to make the information more clear.

I'd suggest also including the numbers identified by the clinician in this table or a separate table that perhaps also includes numbers for TP, FP, FN.

We have added a table to the paper (Table 2 in the revised manuscript) showing the results of the validation test in numbers.

Reviewer 2

Although interesting, the paper does not provide information on the time and effort invested in the development of the adapted system. How was the procedure for developing the final system?

We have explained in the discussion that we have made use of open source plugins available in GATE such as the BioYODIE and SLaM; however we developed our own set of gazetteers and JAPE rules to supplement these plugins and they are available at the GitHub URL specified in the methods section. We have added a line in the 4th paragraph of the methods section documenting the number of gazetteers and JAPE rules used to supplement the existing plugins. We have also added a line in the last paragraph of the weaknesses section that reinforces that human annotation of documents is a time-consuming process.

Which results obtain your final system on the development documents?

The results shown relate to the 200 previously unseen letters which we put through the pipeline in order to validate its performance, we did not include the results from the development documents.

How much code, rules, etc. have been necessary to obtained the final results?

Please see point above

Did you perform any validation on the annotated data?

Yes, Table 2 (Table 3 in the revised manuscript) shows the validation of the annotations produced by the algorithm when compared to those annotated by the clinician. We have also included an additional table (Table 2 in the revised manuscript) showing the differences between the human and the algorithm in terms of false positives, true positives and false negatives.

Did you developed annotation guidelines for the task?

The summary of the guidelines is shown in Table 1. Of note is the way we annotated epilepsy diagnosis, which was based on the stated epilepsy diagnosis, epilepsy syndrome or more than one seizure event, all at a certainty level 4 or above. For medication we extracted only the full prescription annotation, i.e. drug name, quantity, measurement, and a daily dose. No past or future medications were annotated. We have added a line in the caption of table 1 to explain that these definitions were used when developing the algorithm.

Which is the Inter-Tagger Agreement between expert annotators?

We do not provide an inter-annotator agreement as we did not have multiple clinicians available for this study. We have addressed this in the weaknesses section; please see the last paragraph of that section.

Is it far from the system performance?

The “measuring performance” section explains that every disagreement between the human annotator was reviewed to eliminate error in the human annotations – we have added a line in that section explaining this (1st paragraph).

For comparing the new system to previous ones, why not applying your system to discharge letters instead of clinic letters?

The vast majority of patients with epilepsy, unless their disease is very poorly controlled, are seldom admitted to hospital and subsequently discharged, however, they are seen in the hospital outpatient clinics and these visits are documented in the clinic letters. These letters hold detail not available from any other source. Our aim was therefore to develop a system for extracting information from clinic letters not from discharge letters.

Moreover, no error analysis is included. Where the system is still failing? Why? The same problems in both the development and test documents?

Table 2 (Table 3 in the revised manuscript) provides precision, recall, and F1 score for a previously unseen set of 200 letters. During the development of the system we realised that some items were more difficult to extract than others, for example seizure frequency or investigation results, as compared to diagnosis or medication, which we have explained in the first two paragraphs of the discussion. There is much variation in the way this information is written and a lot of ambiguity which is difficult to resolve. We have also provided an additional table (Table 2 in the revised manuscript) that includes the number of false positives, false negatives and true positives to quantify the error across all categories.

Is your system not producing false positives?

Please see Table 2 of the revised manuscript which documents the number of false positives in each category. Also the precision rates indicate the presence of false positives.

No direct comparison on the same data is reported since it seems that there is no public available epilepsy documents. Is it not possible to release the annotated documents?

Although we pseudonymised the letters we are not in a position to release them due to ethical considerations and patient privacy.

How can we establish a fair comparison between existing systems? There are no other available systems for processing your documents? EpiDEA, PEEP, etc. are not available?

We haven't tried other systems to extract epilepsy information from clinic letters and as far as we know there hasn't been one developed to extract specific items of information which we ascertained. Other systems we identified were developed and validated on discharge summaries or investigation reports which represent different writing styles regarding both structure and language. Our system has been developed specifically for clinic letters that generally contain much more unstructured prose compared to discharge letters and investigation reports.

What do you plan as a future work? What and how do you plan to do next?

We are continuing to improve and expand the capability of ExECT and plan to apply it to a larger set of clinic letters.

Minor comments:

There are two possible papers referring to Cui et al. in section "Comparison to other studies" which refers to papers [20] and [21].

Yes, these are two different papers by the same lead author as shown in the references.

VERSION 2 – REVIEW

REVIEWER	SH DC
REVIEW RETURNED	01-Nov-2018

GENERAL COMMENTS	While I appreciate the author's efforts to address concerns in the previous review, I feel there is still some ambiguity as to what elements were assigned a level of certainty. Table 1 in the new Levels of Certainty row says "We applied these certainty levels to each information item associated with the epilepsy diagnosis and epilepsy type," but the example in paragraph 3 of the methods section reads like it is type of seizure. In the author's review response they state, "Levels of certainty were assigned to all diagnostic statements, based on phrases expressing clinical opinion." The authors need to be clear about which categories were assigned certainty levels. If certainty was used with more than one category, I might also suggest that instead of or in addition to having Level of Certainty as it's own row in Table 1, that the authors put asterisks or other symbology with each category that has an associated certainty. Also include text like the, "Only epilepsy diagnosis with a certainty level 4 or 5 was considered to be a true positive," text that currently appears with "Epilepsy diagnosis" in any other category row that also used a certainty. Table 2 has a caption that defines TN=True negative, yet there is no TN column in the table, and it has a column FP with no definition in the caption.
---

VERSION 2 – AUTHOR RESPONSE

Reviewer: 1

Reviewer Name: Saeed Hassanpour

Institution and Country: Dartmouth College

Please state any competing interests or state 'None declared': None

Please leave your comments for the authors below

While I appreciate the author's efforts to address concerns in the previous review, I feel there is still some ambiguity as to what elements were assigned a level of certainty. Table 1 in the new Levels of Certainty row says "We applied these certainty levels to each information item associated with the epilepsy diagnosis and epilepsy type," but the example in paragraph 3 of the methods section reads like it is type of seizure. In the author's review response they state, "Levels of certainty were assigned to all diagnostic statements, based on phrases expressing clinical opinion." The authors need to be clear about which categories were assigned certainty levels. If certainty was used with more than one category, I might also suggest that instead of or in addition to having Level of Certainty as its own row in Table 1, that the authors put asterisks or other symbology with each category that has an associated certainty. Also include text like the, "Only epilepsy diagnosis with a certainty level 4 or 5 was considered to be a true positive," text that currently appears with "Epilepsy diagnosis" in any other category row that also used a certainty.

Thank you, we agree that we should have made it clearer in the manuscript that seizure type forms a part of epilepsy diagnosis. We have clarified this by amending the sentence stating that we applied the certainty levels to "epilepsy diagnosis, epilepsy type, and seizure type" under Levels of Certainty in Table 1. We also added a statement about the certainty levels under Epilepsy Type and Seizure Type rows in the same table; we hope that these changes are adequate.

Table 2 has a caption that defines TN=True negative, yet there is no TN column in the table, and it has a column FP with no definition in the caption.

Thank you for pointing out this error. The definition should have referred to False Positives and has now been corrected.